# Diminishing Noise Maintains Differential Privacy and Enhances Convergence

## Abstract

Differential Privacy (DP) is a well-established framework for training models in distributed settings while safeguarding sensitive information. Although numerous DP algorithms exist, many current solutions inject noise with constant variance to the transmitted gradients, leading to convergence only to a neighborhood of the optimal solution. To address this limitation, we propose an error compensation technique that maintains linear convergence without compromising privacy guarantees. This is achieved through cautious adjusting the noise's variance through the algorithm iterations. Experimental results validate the effectiveness of our approach.

## 1 Introduction

The increasing number of trainable parameters (Villalobos et al., 2022) and dataset sizes lead to more complex machine learning problems that cannot be solved on a single device. One possible solution to this challenge is to distribute the task across several smaller devices and find the solution via local computations and communications (Kairouz et al., 2021). This can be formalized by considering the following optimization problem:

$$\min_{x \in \mathbb{R}^d} \left\{ f(x) = \frac{1}{n} \sum_{i=1}^{n} f_i(x) \right\}.$$

In this context, the function $f(x)$ represents the loss function computed over the entire dataset, while $f_i(x)$ corresponds to its component computed on the $i$-th local device's data.

Federated Learning (FL) has emerged as a prominent distributed learning paradigm where heterogeneous datasets are spread across numerous personal devices. Consequently, this approach faces several challenges. First, there is the requirement for efficient communication. This can be addressed by transmitting compressed gradients (Alistarh et al., 2017), constructing gradient-based sequences to compensate for approximation errors (Karimireddy et al., 2019) or performing several local steps before model averaging (Stich, 2018). These techniques reduce bandwidth consumption while maintaining overall performance. Another key challenge in FL is personalization, i.e., adapting models to local device states. At the same time, the primary objective remains ensuring **privacy** in personalization—enabling collaborative learning while protecting information stored on personal devices. Although FL inherently provides some privacy by exchanging model updates instead of raw data (Duchi et al., 2014), communications may still reveal sensitive information to adversaries (Zhu et al., 2019). To mitigate this, various approaches have been proposed, such as:

- Anonymization (Majeed & Lee, 2020) (which is cheap, but not always effective (Brasher, 2018)),

- Homomorphic encryption (Acar et al., 2018) (which provides fully privacy, but is drastically difficult to implement and expensive to utilize).

Because of this trade-off, new techniques are created and analyzed, that try to include both protectiveness and simplicity. For instance, flipping labels (Shen et al., 2023) or adding noise (Geng & Viswanath, 2015). These mechanisms can be formalized under **Differential Privacy (DP)** (Dwork, 2006), which provides a versatile framework for privacy-preserving methods that are relatively simple to analyze.

## 2 RELATED WORK AND CONTRIBUTION

**Differential Privacy**

The framework of DP (Dwork, 2006; 2008) was developed to address privacy concerns, initially as a formal criterion for dataset security. DP quantifies the risk of an adversary determining whether a specific data point was used in model training or included in a dataset. This framework has been extensively adopted to analyze the privacy guarantees of deep learning algorithms, both in centralized (Bassily et al., 2014; Abadi et al., 2016; Wang et al., 2020; Chen et al., 2021) and distributed settings (McMahan et al., 2018; Andrew et al., 2022; Li et al., 2022).

Its essence lies in privacy budget, which is defined beforehand. Though, we do not achieve fully protection, we determine a certain amount of privacy, which can vary depending on our requirements.

A key development in this field was the DP-SGD algorithm (Abadi et al., 2016), which enforces DP by adding Gaussian noise to gradient updates at each iteration. The noise magnitude required for achieving $(\epsilon, \delta)$-DP depends on the sensitivity of the gradients' magnitude (Dwork, 2006). To bound this sensitivity, DP-SGD employs gradient clipping — a technique now widely used to simplify privacy analysis (Wang et al., 2018; Das et al., 2023).

Though, clipping tend to perform well in modern applications (Zhang et al., 2019), it cooperates poorly with noise injection during training under general assumptions SGD converges only to a neighborhood of the optimum (Koloskova et al., 2023). One may implement adaptive clipping (Andrew et al., 2021; Bu et al., 2023; Pichapati et al., 2019), where threshold diminishes during the training process. However, to our knowledge, existing convergence bounds do not coincide with clipping with the constant radius. Furthermore, in most cases this practice is not theoretically justified.

Recently, novel private algorithms have been introduced (Khirirat et al., 2023; Fatkhullin et al., 2023; Islamov et al., 2025), building on the error-compensating technique (Richtárik et al., 2021). Importantly, this technique was originally developed not for privacy, but for efficient message compression and acceleration in distributed optimization. In the highlighted works, however, the authors replace compression operators with clipping, a standard practice in differential privacy analysis, as noted above.

**Compression**

The demand for efficient communication during the training process lead to development of the schemes, that reduce the number of transmitted bits information (Khirirat et al., 2018). The main idea is to share not exact information (gradients, local states, etc.), but some compressed version of it.

One of the first well-analyzed algorithm for distributed optimization, QSGD (Alistarh et al., 2017), quantized the local gradients. The problem of this approach is in non-reducible variance of the compressed gradients. It can be illustrated as following: in the global optimum the sum of local model's gradients equals to zero, but each of the gradients can be arbitrary large. Therefore, since variance of the quantized vector usually depends on the vector's norm, even at the optimum point, sent gradients can deviate from the real ones by a large margin.

To mitigate this effect, error-compensating approaches were introduced, inspired by the variance reduction methods (Schmidt et al., 2017). The idea is simple. We need to compress not the gradient themselves, but the difference between the gradient and previous estimation. Then, these compressed errors are sent to the server. These approach result in diminishing gradient approximation errors, therefore, their compressed versions are also small.

Initially, these type of methods utilized unbiased compression operators (He et al., 2023), that do not change the compressed vector in expectation. This approach resulted as algorithm `DIANA` (Mishchenko et al., 2019). Later, biased compressors were also bridged with this approach in `EF21` (Richtárik et al., 2021). These days, there are numerous methods that can be considered as error-compensating, for instance, `MARINA` (Gorbunov et al., 2021), `DASHA` (Tyurin & Richtárik, 2022) and `EF-BV` (Condat et al., 2023).

**Our Contribution**

The remarkable performance of error-compensating methods stems from their use of converging-to-zero compressed messages, whose sizes are bounded. Additionally, most DP techniques rely on additive Gaussian noise with constant variance, even though theoretical results suggest noise proportional to the message size should suffice. By connecting these approaches, we develop a

method that leverages the diminishing magnitude of communications to reduce the noise required for privacy preservation.

To summarize all the contribution in comparison with the SOTA results in this field, we:

1. **Propose a new DP version of `EF21` with biased compressors.**
   Previous works employed clipping operators in `EF21`, which facilitated differential privacy analysis but failed to preserve the communication efficiency properties offered by certain biased compressors. We bridge this gap by demonstrating that models can achieve both privacy and bandwidth efficiency during training.

2. **Introduce the concept of diminishing noise in DP setup, achieving the linear convergence.**
   The foundation of our error compensation framework lies in the diminishing magnitude of transmitted messages throughout the optimization process. As these messages converge to zero, their sensitivity similarly decreases, allowing us to employ noise with progressively reducing variance without affecting convergence. We theoretically demonstrate that this diminishing noise preserves linear convergence under the PL condition, matching the convergence rate of the original `EF21` algorithm. Notably, this convergence guarantee was not established in prior works on differentially private error compensation methods.

3. **Validate theoretical results experimentally.**
   We compare our method with existing on `CIFAR-10` dataset, a well-established benchmark for the optimization algorithms.

## 3 PRELIMINARIES AND DEFINITIONS

**Notation.** We use the standard Euclidean norm for vectors: $\|x\| \stackrel{\text{def}}{=} \langle x, x \rangle^{1/2}, x \in \mathbb{R}^d$. The objective functional $f : \mathbb{R}^d \to \mathbb{R}$ is a differentiable function. We denote its global minimum by $f_* \stackrel{\text{def}}{=} \inf_{x \in \mathbb{R}^d} f(x) > -\infty$. We also introduce the gradient of $f$ at point $x$ as $\nabla f(x) \in \mathbb{R}^d$.

Below we introduce all the definitions that will be used throughout the manuscript.

**Definition 1** (Smoothness). *Every $f_i$ has $L_i$-Lipschitz gradient, i.e.*
$$\|\nabla f_i(x) - \nabla f_i(y)\| \le L_i \|x - y\| \quad \forall x, y \in \mathbb{R}^d.$$
*Furthermore, $L^2 \stackrel{\text{def}}{=} \frac{1}{n} \sum_{i=1}^{n} L_i^2$.*

**Definition 2** (PL condition). *There exists $\mu > 0$, such that*
$$f(x) - f_* \le \frac{1}{2\mu} \|\nabla f(x)\|^2 \quad \forall x \in \mathbb{R}^d.$$

These assumptions are standard in stochastic optimization. Smoothness plays a central role in analyzing optimization algorithms in ML: it allows bounding the decrease in loss after each gradient update and enables proper step-size selection (Nesterov, 2013). Regarding PL condition, recently (Liu et al., 2022), it was shown, that over-parametrized neural networks are locally PL almost everywhere, which ensures faster convergence of gradient-based methods. It justifies our analysis, that heavily relies on this property.

All the necessary privacy prerequisites, that we refer to in the analysis, are introduced further:

**Definition 3** (($\varepsilon, \delta$)-DP (Dwork, 2006)). *A randomized mechanism $\mathcal{M} : \mathcal{D} \to \mathcal{R}$ with domain $\mathcal{D}$ and range $\mathcal{R}$ is ($\varepsilon, \delta$)-DP if for all adjacent (different within one element) datasets $D, D' \in \mathcal{D}$ and for all events $S \in \mathcal{R}$ in the output space of $\mathcal{M}$ it holds*
$$\mathbb{P}\{\mathcal{M}(D) \in S\} \le e^\varepsilon \mathbb{P}\{\mathcal{M}(D') \in S\} + \delta.$$

**Definition 4** (Sensitivity). *Function $f : \mathcal{D} \to \mathbb{R}^d$ is said to have sensitivity $\Delta$, if for adjacent datasets $\mathcal{D}' \sim \mathcal{D}$ we have*
$$\Delta^2 = \max_{D \sim D'} \|\mathcal{M}(D) - \mathcal{M}(D')\|^2.$$

**Lemma 1** (Gaussian Mechanism (GM) for DP). *Let $f : \mathcal{X}^n \to \mathbb{R}^d$ has sensitivity $\Delta$. Define a randomized algorithm $\mathcal{M} : \mathcal{X}^n \to \mathbb{R}^d$ by $\mathcal{M}(x) = \mathcal{N}\left(f(x), \frac{2 \log 1.25/\delta}{\varepsilon^2} \Delta^2 I_d\right)$. Then $\mathcal{M}$ is ($\varepsilon, \delta$)-DP.*

**Definition 5** (Privacy Loss). *Let $P$ and $Q$ be two probability distributions on $\mathcal{X}$. Define $f_{P||Q} : \mathcal{X} \to \mathbb{R}$ by $f_{P||Q}(y) = \log \frac{P(y)}{Q(y)}$. The privacy loss is a random variable $PrivLoss(P||Q) \stackrel{def}{=} f_{P||Q}(Y)$, where $Y \sim P$.*

**Definition 6** (Concentrated Differential Privacy). *A randomized mechanism $\mathcal{M} : \mathcal{D} \to \mathcal{R}$ with domain $\mathcal{D}$ and range $\mathcal{R}$ is $\rho$-zCDP if for all adjacent (different within one sample) datasets $D, D' \in \mathcal{D}$ the privacy loss distribution is well-defined and*

$$\mathbb{E}_{Z \sim PrivLoss(\mathcal{M}(D)||\mathcal{M}(D'))} \exp(tZ) \leq \exp\left(t(t+1) \cdot \rho\right), \quad \forall t \geq 0.$$

**Lemma 2** (GM for Concentrated DP). *Let $f : \mathcal{X}^n \to \mathbb{R}^d$ has sensitivity $\Delta$. Define a randomized algorithm $\mathcal{M} : \mathcal{X}^n \to \mathbb{R}^d$ by $\mathcal{M}(x) = \mathcal{N}\left(f(x), \frac{\Delta^2}{2\rho} I_d\right)$. Then $\mathcal{M}$ is $\rho$-zCDP.*

**Lemma 3** (Composition for Concentrated DP). *Let $M_1, M_2, \ldots, M_k : \mathcal{X}^n \to \mathcal{R}$ be randomized algorithms. Suppose, $\mathcal{M}_i$ is $\rho_i$-zCDP for each $i$. Define $\mathcal{M}(x) : \mathcal{X}^n \to \mathcal{R}^k$ by $\mathcal{M}(x) \stackrel{def}{=} (\mathcal{M}_1(x), \mathcal{M}_2(x), \ldots, \mathcal{M}_k(x))$, where each algorithm is run independently. Then, $\mathcal{M}$ is $\rho$-zCDP for $\rho = \sum_{i=1}^{k} \rho_i$.*

**Lemma 4** (Conversion from Concentrated DP to DP). *$\rho$-zCDP implies $\left(\varepsilon = \rho + 2\sqrt{\rho \log 1/\delta}, \delta\right)$-DP for all $\delta > 0$. Also, to obtain a given target $(\varepsilon, \delta)$-DP, it suffices to have $\rho$-zCDP with $\rho \in \left[\frac{\varepsilon^2}{4 \log 1/\delta + 4\varepsilon}, \frac{\varepsilon^2}{4 \log 1/\delta}\right]$*

Let us discuss these claims. DP condition (Definition 3) bounds the probability, that random algorithm differs much on dataset and adjacent one. One of the simplest in implementation technique to provide the privacy is adding Gaussian noise (Lemma 1), which is anisotropic and has variance, proportional to the norm of sensitivity (Definition 4). This is the reason for utilizing clipping in existing DP approaches, since it naturally bounds the sensitivity.

When the adversary has access no only to one message, but instead to some finite numbers of them, he can combine them somehow, therefore we need to protect not the single messages only. It appears, that DP guarantees on single queries may be advanced to its union (Lemma 7).

The problem with Approximate DP is in sophisticated analysis of composition theorems. In previous works (Li et al., 2022; Islamov et al., 2025), they assumed the constant privacy budget per iteration. To consider a more general case, we investigate the framework of concentrated differential privacy (Definition 6), that has not been conducted before in optimization manuscripts. It is similar to approximate DP, as it also is provided via the Gaussian Mechanism (Lemma 6) and is convertible to $(\varepsilon, \delta)$-DP (Lemma 8).

Having discussed the privacy preliminaries we continue to the compression property, as this is the cornerstone of efficient distributed methods, allowing to reduce the communication costs.

Our proposed algorithm relies on biased compressors satisfying the contraction property – a fundamental requirement for convergence in communication-efficient distributed optimization. Unlike unbiased compressors that preserve expectation but may increase the vector's norm drastically, these operators provide controlled compression by consistently maintaining the original vector's direction while bounding the relative magnitude reduction.

**Definition 7** (Biased Compressor). *Mapping $\mathcal{C} : \mathbb{R}^d \to \mathbb{R}^d$ is called a biased compressor, i.e.:*

$$\|C(x) - x\|^2 \leq (1 - \alpha) \|x\|^2, \quad \forall x \in \mathbb{R}^d$$

Some of the most practical biased compressors, are sparsifications, such as `TopK`, and quantizations, as Biased Roundings.

**Example 1** (TopK (Stich et al., 2018)). *Greedy sparsification is defined as*

$$\mathcal{C}(x) = \sum_{i=d-k+1}^{d} x_{(i)} e_{(i)},$$

*where coordinates are ordered by their absolute value, so that $|x_{(1)}| \leq |x_{(2)}| \leq \ldots \leq |x_{(d)}|$ and $e_1, \ldots, e_d$ are a standard Euclidean basis. It satisfies Definition 7 with $\alpha = \frac{k}{d}$.*

**Example 2** (Biased Rounding (Beznosikov et al., 2024)). *Let $\{a_k\}_{k \in \mathbb{Z}}$ be an arbitrary increasing sequence of positive numbers, such that $\inf_k a_k = 0$ and $\sup_k a_k = \infty$. Then, general biased rounding*

*is defined via*

$$\mathcal{C}(x)_i = sign(x_i) \arg \min_{k \in \mathbb{Z}} |a_k - |x_i||, \quad i \in [d].$$

*Then, this operator satisfies Definition 7 with* $1/\alpha = \sup_{k \in \mathbb{Z}} \frac{(a_k + a_{k+1})^2}{4 a_k a_{k+1}}$.

Examples above provide compressors, that allows to reduce the number of transmitted bits. The first one, transmits $k$ coordinates instead of the problem dimension, $d$. Another one is used in the quantization, where we store the model more efficiently.

# 4 MAIN PART

## 4.1 NOISE AFTER COMPRESSION

Having established the necessary background, we can now proceed to the main theoretical contribution of our paper. We introduce the method `DPd-EF21` (Differential Private diminishing `EF21`):

---

**Algorithm 1** `DPd-EF21`

---

1: **Parameters:** starting point $x^0 \in \mathbb{R}^d$, $g_i^0 = 0 \in \mathbb{R}^d$, learning rate $\gamma > 0$.
2: **for** $t = 0, 1, 2, \ldots, T - 1$ **do**
3:      Send $x^t$ to nodes
4:      **for** all nodes $i = 1, 2, \ldots, n$ in parallel **do**
5:          Compress $\Delta_i^t = \mathcal{C}\left(\nabla f_i(x^t) - g_i^{t-1}\right)$
6:          Update local state $g_i^t = g_i^{t-1} + \Delta_i^t$
7:          Send $\Delta_i^t + \mathcal{N}\left(0, \sigma_{i,t}^2 I_d\right)$ to the server
8:      **end for**
9:      Server computes

$$g^t = g^{t-1} + \frac{1}{n} \sum_{i=1}^{n} \left[\Delta_i^t + \mathcal{N}\left(0, \sigma_{i,t}^2 I_d\right)\right] \tag{1}$$

10:      Update the model $x^{t+1} = x^t - \gamma_t g^t$
11: **end for**

---

We utilize the `EF21` approach (Richtárik et al., 2021), transmitting the compressed error between the local estimation and the exact local gradient (line 5). This allows to achieve better convergence, than by transferring gradients, since compressed estimation artifacts tend to converge to zero, that cannot be claimed for the local gradients.

The core idea of the convergence proof is analyzing the Lyapunov function

$$V^t = f(x^t) - f_* + \frac{1}{n} \sum_{i=1}^{n} \|\nabla f_i(x^t) - g_i^{t-1}\|^2. \tag{2}$$

In the original `EF21` paper, Lyapunov equation converge linearly to zero, due to the absense of the stochastic terms:

$$V^t \le (1 - \gamma \mu)^t V^0. \tag{3}$$

Adding a noise with constant variance $\sigma^2$ (line 7) is not helpful in terms of convergence, since we will attract to some neigbourhood, instead of the exact solution (Fatkhullin et al., 2021):

$$\mathbb{E}V^t \le (1 - \gamma \mu)^t \mathbb{E}V^0 + \frac{d\gamma \sigma^2}{\mu}. \tag{4}$$

The novelty we propose is adjusting $\sigma_{i,t}^2$ during the iteration process. To preserve a certain amount on privacy, variance of injected noise should linearly depends on the sensitivity of the mapping we are aiming to protect. In our scenario, this is

$$\Delta_t^2 = \max_{D \sim D'} \left\| \mathcal{C}\left(\nabla f_{\mathcal{D},i}(x^t) - g_{\mathcal{D},i}^{t-1}\right) - \mathcal{C}\left(\nabla f_{\mathcal{D}',i}(x^t) - g_{\mathcal{D}',i}^{t-1}\right) \right\|^2. \tag{5}$$

Since we send $\mathcal{C}\left(\nabla f_i(x^t) - g_i^{t-1}\right)$, we can bound this sensitivity from above by corresponding Lyapunov functions, that depend on their dataset:

$$
\begin{aligned}
\Delta_t^2 &\lesssim \left\|\mathcal{C}\left(\nabla f_{\mathcal{D},i}(x^t) - g_{\mathcal{D},i}^{t-1}\right)\right\|^2 + \max_{\mathcal{D}\sim\mathcal{D}'} \left\|\mathcal{C}\left(\nabla f_{\mathcal{D}',i}(x^t) - g_{\mathcal{D}',i}^{t-1}\right)\right\|^2 \\
&\leq (1-\alpha)\left\|\nabla f_{\mathcal{D},i}(x^t) - g_{\mathcal{D},i}^{t-1}\right\|^2 + (1-\alpha)\max_{\mathcal{D}\sim\mathcal{D}'}\left\|\nabla f_{\mathcal{D}',i}(x^t) - g_{\mathcal{D}',i}^{t-1}\right\|^2 \\
&\lesssim V_{\mathcal{D}}^t + V_{\mathcal{D}_1}^t.
\end{aligned}
$$

As we have $V^t \downarrow$ from the equation (4), we expect $\Delta_t^2$ also to decrease. Since theory proposes, that sensitivity and noise's variance are correlated, we have $\sigma_t^2 \downarrow$. Thus, from the equation (4) we derive $V^t \to 0$ after carefully examining all the terms.

It turns out, that we can apply different strategies in terms of noise adding, depending on the privacy per iteration. This is obtained by advanced composition techniques, that allow various changing DP guarantees per iteration to get a needed privacy overall.

In our analysis, we aim to compound facts above into adding noise with reducing noise in order to maintain the linear convergence in PL scenario. We consider different strategies and end up with the following theorem:

**Theorem 1.** *Let function $f$ be $L$-smooth and satisfy PL condition with constant $\mu$. Define $c_{up} = \frac{4-\alpha}{4-2\alpha}$, $\gamma_{EF}$ as following:*

$$
\gamma_{EF} = \min\left\{\frac{1}{2L\left(1 + \sqrt{\frac{2c_{up}}{8-2\alpha-4c_{up}\alpha}\left(2+\frac{4}{\alpha}\right)}\right)}; \frac{\alpha}{2\mu}\right\}.
$$

*With $T = \Omega\left(\frac{C_1 \cdot C_2}{n}\right)$ there exists a sequence of stepsizes $\{\gamma_k\}_k$, satisfying*

$$
\frac{C_1 \cdot C_2 \cdot \left(1 + L\gamma_k + C_3\gamma_k^2\right)}{n\left(1 - \frac{\gamma_k\mu}{2}\right)(T+1)\exp\left(\frac{(k+1)\gamma_k\mu}{4}\right)} \leq 1, \quad \gamma_k \leq \gamma_{EF}, \quad \gamma_{k-1} \leq \gamma_k \leq c_{up}\gamma_{k-1},
$$

*where $C_1 = C_1(\varepsilon,\delta)$ is the constant, depending on the privacy and $C_2 = C_2(L,\mu,\alpha,d,p)$, as well as $C_3 = C_3(L,\alpha,c_{up})$ – constants, depending on the problem, such that we have following convergence result:*

$$
V^{T+1} \leq 2\prod_{k=0}^{T}\left(1 - \frac{\gamma_k\mu}{4}\right) \cdot \overline{V}^0
$$

*with high probability $1 - p$, where $V^k = f(x^k) - f_* + \frac{2\gamma_k}{\alpha n}\sum_{i=1}^{n}\|\nabla f_i(x^k) - g_i^k\|^2$ and $\overline{V}^0 = \max_{\mathcal{D}'}\left[f(x^0) - f_* + \frac{2\gamma_0}{\alpha n}\sum_{i=1}^{n}\|\nabla f_i(x^0) - g_i^0\|^2\right]$, where maximum is taken across all considered datasets for this problem. Furthermore, Algorithm 1 will be $(\varepsilon,\delta)$-DP.*

We analyze the behaviour of the step sizes $\gamma_k$. After $\gamma_0$ is found, all others stepsizes inevitably exists – one can check, that $\gamma_k$ is eligible for the $k+1$ iteration. It can be noted, that our method have two phases of working: a warming-up DP regime and the main EF21 one, depending on the amount of iterations. Initial step sizes are in the warming-up regime, therefore, small. After several iterations, as we have $\exp(\frac{k\mu\gamma}{4})$ in the denominator, the fraction will be close to zero, hence, we will operate in the main regime with $\gamma \sim \frac{1}{L}$. This is similar to the clipping procedure, where we intentionally reduce the steps. Moreover, as shown in (Khirirat et al., 2023), (Islamov et al., 2025), clipping is not active after a certain amount of iterations, therefore steps are not mitigated. Algorithm 1 behaves the same, where we end up with a constant step size after increasing for the warm-up phase. However, highlighted methods still proceeded to inject noise with constant variance, and not obtaining the exact solution, unlike our proposed framework.

Therefore, the convergence process of our algorithm can be divided into two parts. In the warming up DP regime we will conduct less, than some $T_0$ steps, since the exponent will have the major impact after several iterations. The second regime – the EF one, inherits the linear convergence. Hence, we can derive the number of iterations, required to obtain $\varepsilon$-exact solution.

**Corollary 1.** *To achieve $x^T$ with $f(x^T) - f_* \leq \varepsilon$ with high probability, Algorithm 1 needs*

$$T = T_0 \left( 1 - \frac{\gamma_0}{\gamma_{EF}} \right) + \frac{4}{\gamma_{EF}\mu} \log \frac{\overline{V}^0}{\varepsilon}$$

*iterations.*

It is worth noticing, that our proposed method, unlike other methods, which incorporated differential privacy, converges to exact solution, rather than its neighbourhood.

One of the most important contributions is analyzing the added noise per algorithm step. We derive, that at every iteration the variance of added noise is proportional to $\sim \left( 1 - \frac{\gamma\mu}{2} \right)^t$. It is in fact decreasing, which is crucial for achieving convergence to the exact optimum.

The next highlighted point is is the choice of privacy levels per iteration. Prior approaches employed noise with constant variance throughout training, which resulted in same privacy budget per iteration. In this method, privacy at iteration $t$ is proportional to $\sim \left( 1 - \frac{\gamma\mu}{4} \right)^t \left( 1 - \frac{\gamma\mu}{2} \right)^{-t}$. It can be noted, that the multiplier is greater, than one, therefore, this approach enables stronger information protection during later stages of convergence when model updates contain more valuable information about the optimum, as opposed to early iterations where updates primarily reflect the less informative starting point.

The last, but not least, is the number of iterations. Numerous existing methods (Wang et al., 2017; Li et al., 2022) both under nonconvex and PL assumptions bound the number of conducted iterations $T$, depending on the parameter's properties and privacy budget. We have no upper bound on $T$, but the lower one. This makes sense - we can't achieve adequate convergence with large privacy after one iteration - the the noise term will be too impactful, and not distributed over many iterations. With given privacy budget per iteration, we need at least $\Omega\left(\frac{C_1 \cdot C_2}{n}\right)$ iterations. However, if we vary privacy levels, we can select $T = \Omega\left(\log \frac{C_1 \cdot C_2}{n}\right)$. More details are present in the Appendix.

### 4.2 Noise before Compression

One may argue, that Algorithm 1 is not efficient, since we do not reduce the amount of transmitted information, due to the nature of Gaussian multivariate random vector. Frankly speaking, at line 7 of Algorithm 1 we transmit the uncompressed vector. To overcome this drawback we utilize the post-processing property of DP.

**Lemma 5** (Post-processing). *Suppose $\mathcal{M} : \mathcal{D} \to \mathcal{R}$ is $(\varepsilon, \delta)$-DP ($\rho$-zCDP) and $h : \mathcal{R} \to \mathcal{R}'$ be an arbitrary mapping. Then, $h \circ \mathcal{M}$ is $(\varepsilon, \delta)$-DP ($\rho$-zCDP).*

One cannot compute a function of the output of a private algorithm $\mathcal{M}$ and make it less private. This helps us a lot, since it allows us to utilize compression operators more efficient and to quantize the output of the Gaussian Mechanism. Therefore, we introduce Algorithm `DPd-EF21-2`:

---

**Algorithm 2** `DPd-EF21-2`

---

1: **Parameters:** starting point $x^0 \in \mathbb{R}^d$, $g_i^0 = 0 \in \mathbb{R}^d$, learning rate $\gamma > 0$.
2: **for** $t = 0, 1, 2, \ldots, T - 1$ **do**
3:     Send $x^t$ to nodes
4:     **for** all nodes $i = 1, 2, \ldots, n$ in parallel **do**
5:         Compress $\Delta_i^t = \mathcal{C}\left( \nabla f_i(x^t) - g_i^{t-1} + \mathcal{N}\left(0, \sigma_{i,t}^2 I_d\right) \right)$
6:         Update local state $g_i^t = g_i^{t-1} + \mathcal{C}\left( \nabla f_i(x^t) - g_i^{t-1} \right)$
7:         Send $\Delta_i^t$ to the server
8:     **end for**
9:     Server computes $g^t = g^{t-1} + \frac{1}{n} \sum_{i=1}^n \Delta_i^t$
10:    Update the model $x^{t+1} = x^t - \gamma_t g^t$
11: **end for**

---

Reasoning stays the same, as well, as proof's techniques. Similar to the previous methods we end up with following corollary, concerning convergence process

**Corollary 2.** *Selecting $\gamma_k$ similarly as in Theorem 1, Algorithm 2 requires*

$$T = \mathcal{O}\left(T_0 + \frac{1}{\gamma_{EF}\mu}\log\frac{\overline{V_0}}{\varepsilon}\right)$$

*iterations to achieve $x^T$ with $f(x^T) - f_* \leq \varepsilon$ with high probability.*

One may found, that asymptotically rates in both cases coincide. This justifies and prefers applying compression after adding noise, due to the significantly less consumed bandwidth.

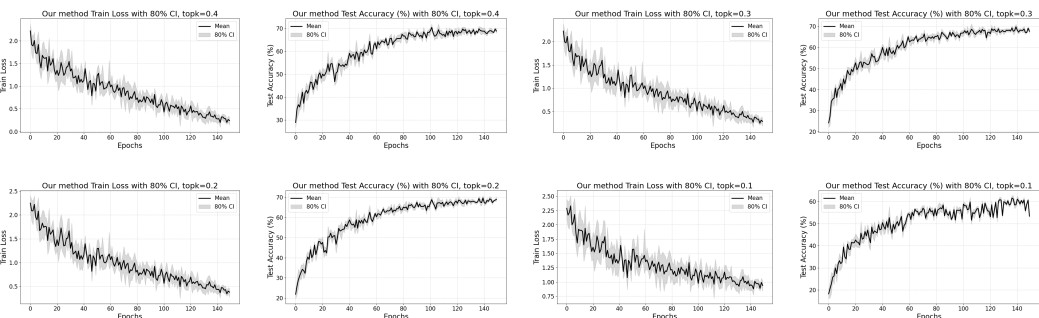

Figure 1: Top-k with varying k

## 5 NUMERICAL EXPERIMENTS

We evaluate our methods on the `CIFAR-10` dataset to study the privacy–utility–communication trade-off. Unless otherwise noted, all methods use the same model and optimizer; full hyperparameters are provided in the appendix. For fairness, all curves are averaged over multiple seeds, and we report the mean with a shaded $\pm 1$ std. band. We also report the realized privacy budget $\varepsilon$ at a fixed $\delta = 10^{-5}$ using the zCDP accountant (Defs. 3–6; Lemmas 2–4), so comparisons are at *matched privacy*. We compare against `DP-Clip21` and `Clip21-SGD2M`.

**Noise schedules.** We consider two diminishing-variance schedules: (i) a geometric schedule $\sigma_t^2 = \left(1 - \frac{\gamma\mu}{4}\right)^t \sigma_0^2$; (ii) a theory-guided schedule with variance proportional to the Lyapunov term, $\sigma_t^2 = \kappa V_t$, where $\kappa$ is chosen so that the per-round zCDP budget composes to the target $(\varepsilon, \delta)$. Both schedules are compatible with our analysis, and we verify them empirically.

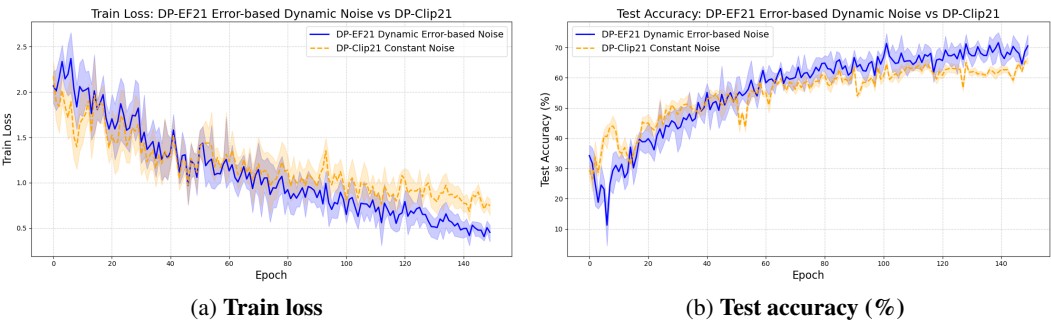

(a) **Train loss**          (b) **Test accuracy (%)**

Figure 2: Our schedule with $\sigma_t^2 \propto V_t$ compared to `DP-Clip21` at matched $(\varepsilon, \delta)$. Diminishing noise attains comparable or better final accuracy and exhibits a lower loss plateau.

We observe that diminishing noise achieves at least the same peak accuracy as `DP-Clip21` and yields a more favorable late-phase plateau in both loss and accuracy.

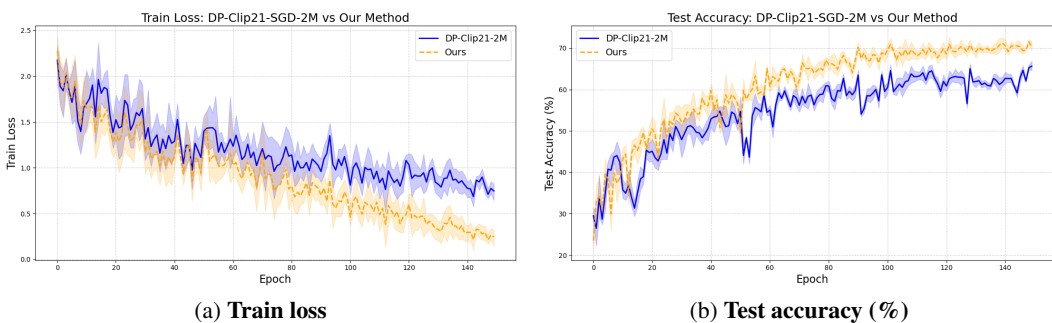

(a) **Train loss**
(b) **Test accuracy (%)**

Figure 3: Comparison to `Clip21-SGD2M` (Islamov et al., 2025) at matched $(\varepsilon, \delta)$. Our diminishing-noise variants are competitive while retaining communication efficiency.

**Communication vs. accuracy.** In practice, different compressors can be employed, and using fewer transmitted coordinates does not necessarily degrade accuracy. We vary the Top-K sparsification level with $k \in \{0.1d, 0.2d, 0.3d, 0.4d\}$ and also explore per-layer $k$. Figure 4 summarizes accuracy as a function of cumulative communicated coordinates (normalized).

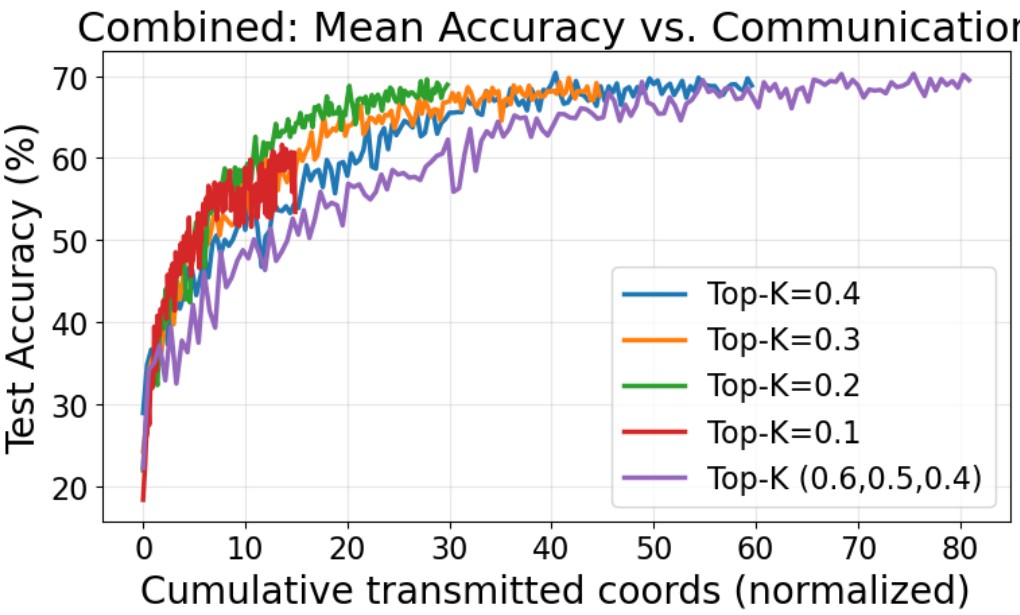

Figure 4: Accuracy vs. cumulative communication under different Top-K levels. Post-processing (noise then compression) preserves privacy while substantially reducing bandwidth.

**Takeaways.** Across matched privacy levels, diminishing-variance noise matches or exceeds the baselines' accuracy while improving late-phase stability, and, when combined with compression after noising, delivers favorable accuracy–communication Pareto fronts.

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

## A  DP STATEMENTS

**Lemma 6** (GM for Concentrated DP). *Let $f : \mathcal{X}^n \to \mathbb{R}^d$ has sensitivity $\Delta$. Define a randomized algorithm $\mathcal{M} : \mathcal{X}^n \to \mathbb{R}^d$ by $\mathcal{M}(x) = \mathcal{N}\left(f(x), \frac{\Delta^2}{2\rho} I_d\right)$. Then $\mathcal{M}$ is $\rho$-zCDP.*

**Lemma 7** (Composition for Concentrated DP). *Let $M_1, M_2, \ldots, M_k : \mathcal{X}^n \to \mathcal{R}$ be randomized algorithms. Suppose, $\mathcal{M}_i$ is $\rho_i$-zCDP for each $i$. Define $\mathcal{M}(x) : \mathcal{X}^n \to \mathcal{R}^k$ by $\mathcal{M}(x) \stackrel{def}{=} (\mathcal{M}_1(x), \mathcal{M}_2(x), \ldots, \mathcal{M}_k(x))$, where each algorithm is run independently. Then, $\mathcal{M}$ is $\rho$-zCDP for $\rho = \sum_{i=1}^{k} \rho_i$.*

**Lemma 8** (Conversion from Concentrated DP to DP). *$\rho$-zCDP implies $\left(\varepsilon = \rho + 2\sqrt{\rho \log 1/\delta}, \delta\right)$-DP for all $\delta > 0$. Also, to obtain a given target $(\varepsilon, \delta)$-DP, it suffices to have $\rho$-zCDP with $\rho = \frac{\varepsilon^2}{4 \log 1/\delta}$*

## B    Descent Lemma

**Lemma 9** (Compressor). *If $\mathcal{C}$ is a biased compressor, then there is a following bound on $g_i^t$, gener-ated by Algorithm 1*

$$\frac{1}{n}\sum_{i=1}^{n}\left\|\nabla f_i(x^{t+1}) - g_i^t\right\|^2 \leq (1 + 2/\alpha)L^2 \left\|x^{t+1} - x^t\right\|^2 + \left(1 - \frac{1}{2/\alpha}\right)\frac{1}{n}\sum_{i=1}^{n}\left\|\nabla f_i(x^t) - g_i^{t-1}\right\|^2$$

*Proof.* Using the Young's inequality, we obtain

$$
\begin{aligned}
\left\|\nabla f_i(x^{t+1}) - g_i^t\right\|^2 &\leq (1 + s)\left\|\nabla f_i(x^{t+1}) - \nabla f_i(x^t)\right\|^2 + \left(1 + s^{-1}\right)\left\|\nabla f_i(x^t) - g_i^t\right\|^2 \\
&\leq (1 + s)L_i^2\|x^{t+1} - x^t\|^2 \\
&+ (1 + s^{-1})\left\|\nabla f_i(x^t) - \mathcal{C}\left(\nabla f_i(x^t) - g_i^{t-1}\right) + g_i^{t-1}\right\|^2 \\
&\leq (1 + \alpha)L_i^2\left\|x^{t+1} - x^t\right\|^2 + \left(1 + s^{-1}\right)(1 - \alpha)\left\|\nabla f_i(x^t) - g_i^{t-1}\right\|^2,
\end{aligned}
$$

which holds $\forall s > 0$. Take $s = 2/\alpha$, hence

$$
\begin{aligned}
\left\|\nabla f_i(x^{t+1}) - g_i^t\right\|^2 &\leq (1 + 2/\alpha)L_i^2\left\|x^{t+1} - x^t\right\|^2 + \left(1 + \frac{1}{2/\alpha}\right)(1 - \alpha)\left\|\nabla f_i(x^t) - g_i^{t-1}\right\|^2 \\
&\leq (1 + 2/\alpha)L_i^2\left\|x^{t+1} - x^t\right\|^2 + \left(1 - \frac{1}{2/\alpha}\right)\left\|\nabla f_i(x^t) - g_i^{t-1}\right\|^2.
\end{aligned}
$$

For the mean deviations we get

$$
\begin{aligned}
\frac{1}{n}\sum_{i=1}^{n}\left\|\nabla f_i(x^{t+1}) - g_i^t\right\|^2 &\leq (1 + 2/\alpha)L^2\left\|x^{t+1} - x^t\right\|^2 \\
&+ \left(1 - \frac{1}{2/\alpha}\right)\frac{1}{n}\sum_{i=1}^{n}\left\|\nabla f_i(x^t) - g_i^{t-1}\right\|^2
\end{aligned}
$$

$\square$

**Lemma 10** (Descent). *Define $V^{t+1} = f(x^{t+1}) - f_* + \theta\gamma_t\frac{1}{n}\sum_{i=1}^{n}\left\|\nabla f_i(x^{t+1}) - g_i^t\right\|^2$, and $w > 0$. Suppose, that $\gamma_{t+1} \leq c_{up}\gamma_t$, where $c_{up} \leq \frac{2M - 2\alpha}{M(2-\alpha)}$ for some $M > \alpha$. Then, with*

$$\gamma_t \leq \gamma_{EF} = \min\left\{\frac{1}{L\left(2 + \sqrt{2\theta c_{up}\left(2 + \frac{4}{\alpha}\right)}\right)}; \frac{2\alpha}{M\mu}\right\}$$

*and $\gamma_t \geq 2^{-1/w}$ Algorithm 1 iterations obtain*

$$
\begin{aligned}
V^{t+1} &\leq \left(1 - \frac{\gamma_t\mu}{2}\right)V^t + n(\gamma_t)\xi_t \\
&= \left(1 - \frac{\gamma_t\mu}{2}\right)V^t + (a\gamma^{1+w} + b\gamma^2 + c\gamma^3)\xi_t,
\end{aligned}
$$

*where $\xi_t \sim \sigma_t^2 \cdot \chi^2(d)$, and $\chi^2(d)$ is a chi-squared random variable with $d$ degrees of freedom, and $a = 1, b = L, c = c_{up}\theta L^2\left(2 + \frac{4}{\alpha}\right), \theta = \frac{M}{2M - M\alpha - 2Mc_{up} + Mc_{up}}.$*

*Proof.* From $L$-smoothness and Young's inequality we obtain:

$$
\begin{aligned}
f(x^{t+1}) - f_* &\leq f(x^t) - f_* + <\nabla f(x^t), x^{t+1} - x^t> + \frac{L}{2}\left\|x^{t+1} - x^t\right\|^2 \\
&= f(x^t) - f_* - \gamma_t \left\langle \nabla f(x^t), g^t_{EF} + g^t_N \right\rangle + \frac{L\gamma_t^2}{2}\left\|g^t_{EF} + g^t_N\right\|^2 \\
&\leq f(x^t) - f_* - \gamma_t \left\langle \nabla f(x^t), g^t_{EF} + g^t_N \right\rangle + L\gamma_t^2\left\|g^t_{EF}\right\|^2 + L\gamma_t^2\left\|g^t_N\right\|^2 \\
&= f(x^t) - f_* - \gamma_t \left\langle \nabla f(x^t), g^t_{EF} \right\rangle - \gamma_t \left\langle \nabla f(x^t), g^t_N \right\rangle \\
&\quad + L_t\gamma^2\left\|g^t_{EF}\right\|^2 + L\gamma_t^2\left\|g^t_N\right\|^2 \\
&= f(x^t) - f_* - \frac{\gamma_t}{2}\left\|\nabla f(x^t)\right\|^2 + \frac{\gamma_t}{2}\left\|\nabla f(x^t) - g^t_{EF}\right\|^2 \\
&\quad - \gamma_t \left\langle \nabla f(x^t), g^t_N \right\rangle + L\gamma_t^2\left\|g^t_{EF}\right\|^2 + \left(L\gamma_t^2 - \frac{\gamma_t}{2}\right)\left\|g^t_N\right\|^2 \\
&\leq f(x^t) - f_* - \frac{\gamma_t}{2}\left\|\nabla f(x^t)\right\|^2 + \frac{\gamma_t}{2}\left\|\nabla f(x^t) - g^t_{EF}\right\|^2 \\
&\quad + \frac{\gamma_t}{4}\left\|\nabla f(x^t)\right\|^2 + \gamma_t\left\|g^t_N\right\|^2 + \left(L\gamma_t^2 - \frac{\gamma_t}{2}\right)\left\|g^t_{EF}\right\|^2 + L\gamma_t^2\left\|g^t_{EF}\right\|^2 \\
&\leq \left(1 - \frac{\gamma_t\mu}{2}\right)\left(f(x^t) - f_*\right) + \frac{\gamma_t}{2}\left\|\nabla f(x^t) - g^t_{EF}\right\|^2 + \left(L\gamma_t^2 - \frac{\gamma_t}{2}\right)\left\|g^t_{EF}\right\|^2 \\
&\quad + \left(\gamma_t + L\gamma_t^2\right)\left\|g^t_N\right\|^2 .
\end{aligned}
$$

From Lemma 9 we have

$$
\begin{aligned}
\frac{1}{n}\sum_{i=1}^n \left\|\nabla f_i(x^{t+1}) - g^t_i\right\|^2 &\leq (1 + 2/\alpha)L^2\left\|x^{t+1} - x^t\right\|^2 \\
&\quad + \left(1 - \frac{1}{2/\alpha}\right)\frac{1}{n}\sum_{i=1}^n \left\|\nabla f_i(x^t) - g^{t-1}_i\right\|^2 \\
&= (1 + 2/\alpha)\gamma_t^2 L^2\left\|g^t_{EF} + g^t_N\right\|^2 \\
&\quad + \left(1 - \frac{1}{2/\alpha}\right)\frac{1}{n}\sum_{i=1}^n \left\|\nabla f_i(x^t) - g^{t-1}_i\right\|^2 \\
&\leq (2 + 4/\alpha)\gamma_t^2 L^2\left\|g^t_{EF}\right\|^2 + (2 + 4/\alpha)\gamma^2 L^2\left\|g^t_N\right\|^2 \\
&\quad + \left(1 - \frac{1}{2/\alpha}\right)\frac{1}{n}\sum_{i=1}^n \left\|\nabla f_i(x^t) - g^{t-1}_i\right\|^2
\end{aligned}
$$

Since $g^t_{EF} = \frac{1}{n}\sum_{i=1}^n g^t_i$, $\nabla f(x^t) = \frac{1}{n}\sum_{i=1}^n \nabla f_i(x^t)$ from convexity we achieve

$$
\left\|\nabla f(x^t) - g^{t-1}_{EF}\right\|^2 = \left\|\frac{1}{n}\sum_{i=1}^n \nabla f_i(x^t) - g^{t-1}_i\right\|^2 \leq \frac{1}{n}\sum_{i=1}^n \left\|\nabla f_i(x^t) - g^{t-1}_i\right\|^2 .
$$

Define $V^{t+1} = f(x^{t+1}) - f_* + \theta\gamma_t \frac{1}{n}\sum_{i=1}^{n}\left\|\nabla f_i(x^{t+1}) - g_i^t\right\|^2$, where $\theta > 0$ and will be declared further. Therefore, we have

$$
\begin{aligned}
V^{t+1} \leq\ & \left(1 - \frac{\gamma_t\mu}{2}\right)\left(f(x^t) - f_*\right) + \frac{\gamma_t}{2}\left\|\nabla f(x^t) - g_{EF}^t\right\|^2 + \left(L\gamma_t^2 - \frac{\gamma_t}{2}\right)\left\|g_{EF}^t\right\|^2 \\
& + \left(\gamma_t + L\gamma_t^2\right)\left\|g_N^t\right\|^2 + (2 + 4/\alpha)\theta\gamma_t^2\gamma_{t+1}L^2\left\|g_{EF}^t\right\|^2 + (2 + 4/\alpha)\theta\gamma_t^2\gamma_{t+1}L^2\left\|g_N^t\right\|^2 \\
& + \left(1 - \frac{1}{2/\alpha}\right)\theta\gamma_{t+1}\frac{1}{n}\sum_{i=1}^{n}\left\|\nabla f_i(x^t) - g_i^{t-1}\right\|^2 \\
\leq\ & \left(1 - \frac{\gamma_t\mu}{2}\right)\left(f(x^t) - f_*\right) + \left(\frac{\gamma_t}{2} + \theta\gamma_{t+1}\left(1 - \frac{1}{2/\alpha}\right)\right)\sum_{i=1}^{n}\left\|\nabla f_i(x^t) - g_i^{t-1}\right\|^2 \\
& + \left(L\gamma_t^2 - \frac{\gamma_t}{2} + (2 + 4/\alpha)\theta\gamma_t^2\gamma_{t+1}L^2\right)\left\|g_{EF}^t\right\|^2 + \left(\gamma_t + \gamma_t^2(L + L^2(2 + 4/\alpha)\theta\gamma_{t+1})\right)\left\|g_N^t\right\|^2 .
\end{aligned}
$$

With $\theta = \frac{M}{2M - M\alpha - 2Mc_{up} + Mc_{up}}$

$$
\begin{aligned}
V^{t+1} \leq\ & \left(1 - \frac{\gamma_t\mu}{2}\right)\left(f(x^t) - f_*\right) + \theta\gamma_t\left(1 - \frac{\alpha}{M}\right)\frac{1}{n}\sum_{i=1}^{n}\left\|\nabla f_i(x^t) - g_i^{t-1}\right\|^2 \\
& + \left(L\gamma_t^2 - \frac{\gamma_t}{2} + (2 + 4/\alpha)\theta c_{up}\gamma_t^3 L^2\right)\left\|g_{EF}^t\right\|^2 + \left(\gamma_t + \gamma_t^2\left(L + L^2(2 + 4/\alpha)\theta c_{up}\gamma_t\right)\right)\left\|g_N^t\right\|^2 .
\end{aligned}
$$

For $\gamma_{EF}$, defined as following:

$$
\gamma_{EF} = \min\left\{\frac{1}{L\left(2 + \sqrt{2c_{up}\theta\left(2 + \frac{4}{\alpha}\right)}\right)}; \frac{2\alpha}{M\mu}\right\},
$$

for every $\gamma_t \leq \gamma_{EF}$ we have

$$
V^{t+1} \leq \left(1 - \frac{\gamma_t\mu}{2}\right)V^t + \left(\gamma_t + \gamma_t^2\left(L + L^2(2 + 4/\alpha)\theta c_{up}\gamma_t\right)\right)\left\|g_N^t\right\|^2 .
$$

as $g_N \sim \mathcal{N}\left(0, \sigma_t^2 \cdot I\right)$, we obtain the needed. $\qquad\square$

**Corollary 3.** *With $M = 4$ we get $\theta = \frac{4}{8 - 2\alpha - 4c_{up}\alpha}$, $c_{up} \leq \frac{4-\alpha}{4-2\alpha}$,*

$$
\gamma_{EF} = \min\left\{\frac{1}{2L\left(1 + \sqrt{\frac{2c_{up}}{8 - 2\alpha - 4c_{up}\alpha}\left(2 + \frac{4}{\alpha}\right)}\right)}; \frac{\alpha}{2\mu}\right\}
$$

## C    MAIN THEOREM

**Theorem 2.** *Suppose, that $V^t$ is a Lyapunov function, defined by*

$$V^t = f(x^t) - f_* + \frac{\theta\gamma_t}{n} \sum_{i=1}^{n} \|\nabla f_i(x^t) - g^{t-1}\|^2,$$

*and let $\sigma_{i,t}^2$ be a variance of DP noise, injected on device $i$. Then, with proper choice of $\gamma_t \geq \gamma_{t-1}$ we might achieve linear convergence for $V^t$ on any dataset:*

$$V^t(\mathcal{D}) \leq 2 \prod_{k=0}^{t-1} \left(1 - \frac{\gamma_k\mu}{4}\right) \cdot \overline{V}^0 \quad \forall \mathcal{D},$$

*where $\overline{V}^0 = \max_{\mathcal{D}'} V^0(\mathcal{D}')$.*

*Proof.* We will proof this by induction. Base is true:

$$V^0(\mathcal{D}) \leq 2 \max_{\mathcal{D}'} V^0(\mathcal{D}')$$

Then we prove bound on sensitivity, We have

$$
\begin{aligned}
\Delta_{i,t}^2 &= \max_{\mathcal{D}_i \sim \mathcal{D}_i'} \left\| \mathcal{C}\left(\nabla f_i(x^t) - g_i^{t-1}\right) - \mathcal{C}\left(\nabla \hat{f}_i(x^t) - \hat{g}_i^{t-1}\right) \right\|^2 \\
&\leq 2 \left\| \mathcal{C}\left(\nabla f_i(x^t) - g_i^{t-1}\right) \right\|^2 + 2 \max_{\mathcal{D}_i \sim \mathcal{D}_i'} \left\| \mathcal{C}\left(\nabla \hat{f}_i(x^t) - \hat{g}_i^{t-1}\right) \right\|^2 \\
&\leq 2(1+\alpha) \left\| \mathcal{C}\left(\nabla f_i(x^t) - g_i^{t-1}\right) - \left(\nabla f_i(x^t) - g_i^{t-1}\right) \right\|^2 \\
&+ 2(1+\alpha^{-1}) \left\| \nabla f_i(x^t) - g_i^{t-1} \right\|^2 \\
&+ \max_{\mathcal{D}_i \sim \mathcal{D}_i'} \left[ 2(1+\beta) \left\| \mathcal{C}\left(\nabla \hat{f}_i(x^t) - \hat{g}_i^{t-1}\right) - \left(\nabla \hat{f}_i(x^t) - \hat{g}_i^{t-1}\right) \right\|^2 \right. \\
&+ 2(1+\beta^{-1}) \left\| \nabla \hat{f}_i(x^t) - \hat{g}_i^{t-1} \right\|^2 \Big] \\
&\leq 2 \left((1+\alpha)(1-\alpha) + (1+\alpha^{-1})\right) \left\| \nabla f_i(x^t) - g_i^{t-1} \right\|^2 \\
&+ 2 \left((1+\beta)(1-\alpha) + (1+\beta^{-1})\right) \max_{\mathcal{D}_i \sim \mathcal{D}_i'} \left\| \nabla \hat{f}_i(x^t) - \hat{g}_i^{t-1} \right\|^2
\end{aligned}
$$

Take $\alpha = \beta = (1-\alpha)^{-1/2}$, then

$$\Delta_{i,t}^2 \leq 2 \left(1 + \sqrt{1-\alpha}\right)^2 \left( \left\| \nabla f_i(x^t) - g_i^{t-1} \right\|^2 + \max_{\mathcal{D}_i \sim \mathcal{D}_i'} \left\| \nabla \hat{f}_i(x^t) - \hat{g}_i^{t-1} \right\|^2 \right).$$

One can notice, that $\frac{1}{n} \sum_{i=1}^{n} \left\| \nabla f_i(x^t) - g_i^{t-1} \right\|^2 \leq \frac{1}{\theta\gamma_t} V^t$. Then,

$$\frac{1}{n} \sum_{i=1}^{n} \Delta_{i,t}^2 \leq \frac{2}{\theta\gamma_t} \left(1 + \sqrt{1-\alpha}\right)^2 \left( V^t(\mathcal{D}) + \frac{1}{n} \sum_{i=1}^{n} \max_{\mathcal{D}_i \sim \mathcal{D}_i'} \left\| \nabla \hat{f}_i(x^t) - \hat{g}_i^{t-1} \right\|^2 \right).$$

As choice of datasets $\mathcal{D}_i'$ depends the device $i$, bound is less strict: $\left\| \nabla \hat{f}_i(x^t) - \hat{g}_i^{t-1} \right\|^2 \leq \frac{n}{\theta} V^t(\mathcal{D}')$. Therefore, applying the induction presumption we have

$$\frac{1}{n} \sum_{i=1}^{n} \Delta_{i,t}^2 \leq \frac{4}{\theta\gamma_t} \left(1 + \sqrt{1-\alpha}\right)^2 \left( V^t(\mathcal{D}) + \sum_{i=1}^{n} V^t(\mathcal{D}') \right) \leq \frac{A}{\gamma_t} \prod_{k=0}^{t-1} \left(1 - \frac{\gamma_k\mu}{4}\right) \cdot \overline{V}^0,$$

where $A = \frac{4(n+1)}{\theta}\left(1+\sqrt{1-\alpha}\right)^2$. Then we unroll the inequalities from the descent lemma:

$$
\begin{aligned}
V^{t+1} &\leq \left(1-\frac{\gamma_t\mu}{2}\right)V^t + n(\gamma_t)\xi_t \leq \prod_{k=0}^{t}\left(1-\frac{\gamma_k\mu}{2}\right)\cdot V^0 + \sum_{k=0}^{t}n(\gamma_k)\xi_k\prod_{i=k+1}^{t}\left(1-\frac{\gamma_i\mu}{2}\right) \\
&\leq \prod_{k=0}^{t}\left(1-\frac{\gamma_k\mu}{2}\right)\cdot \overline{V}^0 + \sum_{k=0}^{t}n(\gamma_k)\xi_k\prod_{i=k+1}^{t}\left(1-\frac{\gamma_i\mu}{2}\right).
\end{aligned}
$$

For $\xi_t$ we have following concentration inequality:

$$
\mathbb{P}\left[\xi_t \geq \sigma_t^2 d(1+\varepsilon_t)\right] \leq \exp\left(-\frac{d}{4}\min(\varepsilon_t, \varepsilon_t^2)\right).
$$

Define $\varepsilon_k = \max\{1, \beta(k+1)\}$, where $\beta$ will be determined further. Therefore, always $\min\{\varepsilon_k, \varepsilon_k^2\} = \varepsilon_k$. Then,

$$
\begin{aligned}
\varepsilon_k &\geq \beta(k+1), \\
-\varepsilon_k &\leq -\beta(k+1), \\
\exp\left(-\frac{d\varepsilon_k}{4}\right) &\leq \exp\left(-\frac{d\beta(k+1)}{4}\right)
\end{aligned}
$$

and

$$
\sum_{k=0}^{t}\exp\left(-\frac{d\varepsilon_k}{4}\right) \leq \sum_{k=0}^{t}\exp\left(-\frac{d\beta(k+1)}{4}\right) \leq \frac{\exp(-d\beta/4)}{1-\exp(-d\beta/4)}.
$$

To bound this with $p$ we need to take

$$
\beta = \frac{4}{d}\ln\left(1+\frac{1}{p}\right)
$$

Therefore, with probability at least $1-p$ we have

$$
V^{t+1} \leq \prod_{k=0}^{t}\left(1-\frac{\gamma_k\mu}{2}\right)\cdot\overline{V}^0 + d\sum_{k=0}^{t}n(\gamma_k)\sigma_k^2(1+\varepsilon_k)\prod_{i=k+1}^{t}\left(1-\frac{\gamma_i\mu}{2}\right)
$$

Next, using this derived inequality we will derive the overall convergence. As shown earlier, if iteration $k$ satisfies $\rho_k$-zCDP for device $i$, if added Gaussian noise have variance $\sigma_{i,k}^2 = \frac{\Delta_{i,k}^2}{2\rho_k}$. If every iteration is $\rho_k$-zCDP, then, adaptive composition will be $\sum_{k=0}^{t}\rho_k$-zCDP. Then, $\rho_k \stackrel{\text{def}}{=} \frac{\nu_k}{Z}\frac{\varepsilon^2}{4\log 1/\delta}$, where $Z = \sum_{k=0}^{t}\nu_k$. Then, according to previous lemma, adaptive composition will be $(\varepsilon, \delta)$-DP.

Note, that we analyzed $\sigma_{i,t}$ instead of $\sigma_t$. However, we have $\mathcal{N}(0, \sigma_t^2 I) = \frac{1}{n}\sum_{i=1}^{n}\mathcal{N}(0, \sigma_{i,t}^2 I)$. Hence, $\sigma_t^2 = \frac{1}{n^2}\sum_{i=1}^{n}\sigma_{i,t}^2$. Therefore, $\sigma_t^2 = \frac{1}{2\rho_t n^2}\sum_{i=1}^{n}\Delta_{i,t}^2 \leq \frac{1}{2\rho_t}\frac{A}{n\gamma_t}\prod_{k=0}^{t}\left(1-\frac{\gamma_k\mu}{4}\right)\overline{V}^0$. Therefore,

$$
\begin{aligned}
V^{t+1} &\leq \prod_{k=0}^{t}\left(1-\frac{\gamma_k\mu}{2}\right)\cdot\overline{V}^0 + \frac{Ad\overline{V}^0}{2n}\sum_{k=0}^{t}\frac{n(\gamma_k)}{\gamma_k\rho_k}(1+\varepsilon_k)\prod_{i=0}^{k-1}\left(1-\frac{\gamma_i\mu}{4}\right)\cdot\prod_{i=k+1}^{t}\left(1-\frac{\gamma_i\mu}{2}\right) \\
&\leq \prod_{k=0}^{t}\left(1-\frac{\gamma_k\mu}{2}\right)\cdot\overline{V}^0 + \frac{Ad\overline{V}^0}{2n}\sum_{k=0}^{t}\frac{n(\gamma_k)}{\gamma_k\rho_k}(1+\varepsilon_k)\prod_{i=0}^{k-1}\left(1-\frac{\gamma_i\mu}{4}\right)\cdot\prod_{i=k}^{t-1}\left(1-\frac{\gamma_i\mu}{2}\right) \\
&= \prod_{k=0}^{t}\left(1-\frac{\gamma_k\mu}{2}\right)\cdot\overline{V}^0 + \frac{Ad\overline{V}^0}{2n}\sum_{k=0}^{t}\frac{n(\gamma_k)}{\gamma_k\rho_k}(1+\varepsilon_k)\prod_{i=0}^{k-1}\left(1-\frac{\gamma_i\mu}{4}\right)\left(1-\frac{\gamma_i\mu}{2}\right)^{-1}\cdot\prod_{i=0}^{t-1}\left(1-\frac{\gamma_i\mu}{2}\right) \\
&= \prod_{k=0}^{t}\left(1-\frac{\gamma_k\mu}{2}\right)\cdot\overline{V}^0\cdot\left[1+\frac{Ad/(2n)}{1-\gamma_t\mu/2}\sum_{k=0}^{t}\frac{n(\gamma_k)}{\gamma_k\rho_k}(1+\varepsilon_k)\prod_{i=0}^{k-1}\left(\frac{1-\frac{\gamma_i\mu}{4}}{1-\frac{\gamma_i\mu}{2}}\right)\right].
\end{aligned}
$$

Let $h(y) = \frac{1-y/4}{1-y/2}$. We have $h'(y) = \frac{1}{(y-2)^2}$, therefore, $h$ is increasing. Define $\nu_k = \prod_{i=0}^{k-1} \left( \frac{1-\frac{\gamma_i \mu}{4}}{1-\frac{\gamma_i \mu}{2}} \right)^{-1} \cdot \left( \frac{1-\frac{\gamma_{EF}\mu}{2}}{1-\frac{\gamma_{EF}\mu}{4}} \right)^k$. Then,

$$Z = \sum_{k=0}^{t} \prod_{i=0}^{k-1} \left( \frac{1-\frac{\gamma_i\mu}{4}}{1-\frac{\gamma_i\mu}{2}} \right)^{-1} \cdot \left( \frac{1-\frac{\gamma_{EF}\mu}{2}}{1-\frac{\gamma_{EF}\mu}{4}} \right)^k \geq \sum_{k=0}^{t} \left( \frac{1-\frac{\gamma_{EF}\mu}{4}}{1-\frac{\gamma_{EF}\mu}{2}} \right)^{-k} \cdot \left( \frac{1-\frac{\gamma_{EF}\mu}{2}}{1-\frac{\gamma_{EF}\mu}{4}} \right)^k = t+1$$

and

$$\frac{1}{Z} \leq \frac{1}{t+1}$$

So, we finally obtain

$$V^{t+1} \leq \prod_{k=0}^{t} \left( 1 - \frac{\gamma_k\mu}{2} \right) \cdot \overline{V}^0 \cdot \left[ 1 + \frac{Ad/(2n)}{1-\gamma_t\mu/2} \frac{1}{t+1} \frac{\varepsilon^2}{4\log 1/\delta} \sum_{k=0}^{t} \frac{n(\gamma_k)}{\gamma_k} \left( \frac{1-\frac{\gamma_{EF}\mu}{2}}{1-\frac{\gamma_{EF}\mu}{4}} \right)^k (1+\varepsilon_k) \right]$$

Use the fact, that $\frac{n(\gamma_k)}{\gamma_k} = \frac{a\gamma_k + b\gamma_k^2 + c\gamma_k^3}{\gamma_k} = a + b\gamma_k + c\gamma_k^2 = m(\gamma_k) \leq m(\gamma_t)$ for $k \leq t$. Then,

$$V^{t+1} \leq \prod_{k=0}^{t} \left( 1 - \frac{\gamma_k\mu}{2} \right) \cdot \overline{V}^0 \cdot \left[ 1 + \frac{Ad/(2n)}{1-\gamma_t\mu/2} \frac{1}{t+1} \frac{\epsilon^2}{4\log 1/\delta} m(\gamma_t) \sum_{k=0}^{t} \left( \frac{1-\frac{\gamma_{EF}\mu}{2}}{1-\frac{\gamma_{EF}\mu}{4}} \right)^k (1+\epsilon_k) \right]$$

$$\leq \prod_{k=0}^{t} \left( 1 - \frac{\gamma_k\mu}{2} \right) \cdot \overline{V}^0 \cdot \left[ 1 + \frac{Ad/(2n)}{1-\gamma_t\mu/2} \frac{1}{t+1} \frac{\epsilon^2}{4\log 1/\delta} m(\gamma_t) \left( 2\sum_{k=0}^{\infty} \left( \frac{1-\frac{\gamma_{EF}\mu}{2}}{1-\frac{\gamma_{EF}\mu}{4}} \right)^k + \beta \sum_{k=0}^{t} k \left( \frac{1-\frac{\gamma_{EF}\mu}{2}}{1-\frac{\gamma_{EF}\mu}{4}} \right)^k \right) \right]$$

$$\leq \prod_{k=0}^{t} \left( 1 - \frac{\gamma_k\mu}{2} \right) \cdot \overline{V}^0 \cdot \left[ 1 + \frac{Ad/(2n)}{1-\gamma_t\mu/2} \frac{1}{t+1} \frac{\epsilon^2}{4\log 1/\delta} m(\gamma_t) \left( 2\sum_{k=0}^{\infty} \left( \frac{1-\frac{\gamma_{EF}\mu}{2}}{1-\frac{\gamma_{EF}\mu}{4}} \right)^k + \beta \sum_{k=0}^{\infty} k \left( \frac{1-\frac{\gamma_{EF}\mu}{2}}{1-\frac{\gamma_{EF}\mu}{4}} \right)^k \right) \right]$$

$$\leq \prod_{k=0}^{t} \left( 1 - \frac{\gamma_k\mu}{2} \right) \cdot \overline{V}^0 \cdot \left[ 1 + \frac{Ad/(2n)}{1-\gamma_t\mu/2} \frac{1}{t+1} \frac{\epsilon^2}{4\log 1/\delta} m(\gamma_t) \left( \frac{2}{1-\frac{1-\frac{\gamma_{EF}\mu}{2}}{1-\frac{\gamma_{EF}\mu}{4}}} + \frac{\beta \cdot \frac{1-\frac{\gamma_{EF}\mu}{2}}{1-\frac{\gamma_{EF}\mu}{4}}}{\left(1-\frac{1-\frac{\gamma_{EF}\mu}{2}}{1-\frac{\gamma_{EF}\mu}{4}}\right)^2} \right) \right]$$

$$= \prod_{k=0}^{t} \left( 1 - \frac{\gamma_k\mu}{2} \right) \cdot \overline{V}^0 \cdot \left[ 1 + \frac{Ad/(2n)}{1-\gamma_t\mu/2} \frac{1}{t+1} \frac{\epsilon^2}{4\log 1/\delta} m(\gamma_t) \left( \frac{8\left(1-\frac{\gamma_{EF}\mu}{4}\right)}{\gamma_{EF}\mu} + \frac{16\beta\left(1-\frac{\gamma_{EF}\mu}{2}\right)\left(1-\frac{\gamma_{EF}\mu}{4}\right)}{(\gamma_{EF}\mu)^2} \right) \right]$$

Overall, we have following situation:

$$V^{t+1} \leq \prod_{k=0}^{t} \left( 1 - \frac{\gamma_k\mu}{2} \right) \cdot \overline{V}^0 \cdot \mathrm{func}(\gamma_t).$$

We want

$$V^{t+1} \leq 2 \prod_{k=0}^{t} \left( 1 - \frac{\gamma_k\mu}{4} \right) \cdot \overline{V}^0,$$

therefore, we need

$$\mathrm{func}(\gamma_t) \leq 2 \prod_{k=0}^{t} \left( \frac{1-\frac{\gamma_k\mu}{4}}{1-\frac{\gamma_k\mu}{2}} \right).$$

We have

$$\frac{1-\frac{\gamma_k\mu}{4}}{1-\frac{\gamma_k\mu}{2}} \geq \frac{1-\frac{\gamma_t\mu}{4}}{1-\frac{\gamma_t\mu}{2}},$$

therefore, it is sufficient to proof

$$\mathrm{func}(\gamma_t) \leq 2 \left( \frac{1-\frac{\gamma_t\mu}{4}}{1-\frac{\gamma_t\mu}{2}} \right)^{t+1} = 2 \left( \frac{1-\frac{\gamma_t\mu}{2}}{1-\frac{\gamma_t\mu}{4}} \right)^{-t-1} = 2 \left( 1 - \frac{\frac{\gamma_t\mu}{4}}{1-\frac{\gamma_t\mu}{4}} \right)^{-t-1},$$

or

$$\mathrm{func}(\gamma_t) \left( 1 - \frac{\frac{\gamma_t\mu}{4}}{1-\frac{\gamma_t\mu}{4}} \right)^{t+1} \leq 2.$$

As we have $(1-x)^k \leq e^{-kx}$, for $x < 1$ we obtain

$$\text{func}(\gamma_t) \left(1 - \frac{\frac{\gamma_t \mu}{4}}{1 - \frac{\gamma_t \mu}{4}}\right)^{t+1} \leq \text{func}(\gamma_t) \exp\left(-(t+1)\frac{\gamma_t \mu}{4 - \gamma_t \mu}\right) \leq \text{func}(\gamma_t) \exp\left(-\frac{(t+1)\gamma_t \mu}{4}\right).$$

Finally, it is sufficient to guarantee, that

$$\frac{\text{func}(\gamma_t)}{\exp\left(\frac{(t+1)\gamma_t \mu}{4}\right)} \leq 2$$

Putting all the constants together we obtain the needed. $\square$

**Remark 1.** *We can take non-decreasing $\gamma_y$, since if $\gamma_t$ satisfies the constraints and iteration $t$, $\gamma_t$ will satisfy constraints at iteration $t + 1$ too. Due to the inevitable increase of the exponent, $\gamma_{EF}$ will be eligible at some point, therefore, we will end up with linear convergence.*

## D    VARYING PRIVACY LEVELS

One can notice, that careful choice of constant $q$ will directly influence the privacy levels per iteration.

Define $\nu_k = \prod_{i=0}^{k-1} \left( \frac{1-\frac{\gamma_i \mu}{4}}{1-\frac{\gamma_i \mu}{2}} \right)^{-1} \cdot q^{-k}$, where $q \in \left( \frac{1-\frac{\gamma_{EF}\mu}{2}}{1-\frac{\gamma_{EF}}{4}}; 1 \right)$. Then,

$$Z = \sum_{k=0}^{t} \prod_{i=0}^{k-1} \left( \frac{1-\frac{\gamma_{EF}\mu}{4}}{1-\frac{\gamma_i \mu}{2}} \right)^{-1} \cdot q^{-k} \geq \sum_{k=0}^{t} \left( \frac{1-\frac{\gamma_{EF}\mu}{4}}{1-\frac{\gamma_{EF}\mu}{2}} \right)^{-k} \cdot q^{-k} = \frac{1 - \left( \frac{1-\frac{\gamma_{EF}\mu}{4}}{1-\frac{\gamma_{EF}\mu}{2}} \right)^{-t-1} q^{-t-1}}{1 - \left( \frac{1-\frac{\gamma_{EF}\mu}{4}}{1-\frac{\gamma_{EF}\mu}{2}} \right)^{-1} q^{-1}}$$

and

$$\frac{1}{Z} \leq \frac{1 - \left( \frac{1-\frac{\gamma_{EF}\mu}{4}}{1-\frac{\gamma_{EF}\mu}{2}} \right)^{-1} q^{-1}}{1 - \left( \frac{1-\frac{\gamma_{EF}\mu}{4}}{1-\frac{\gamma_{EF}\mu}{2}} \right)^{-t-1} q^{-t-1}}$$

So, we finally obtain

$$V^{t+1} \leq \prod_{k=0}^{t} \left( 1 - \frac{\gamma_k \mu}{2} \right) \cdot \overline{V}^0 \cdot \left[ 1 + \frac{Ad/(2n)}{1 - \gamma_t \mu/2} \frac{1 - \left( \frac{1-\frac{\gamma_{EF}\mu}{4}}{1-\frac{\gamma_{EF}\mu}{2}} \right)^{-1} q^{-1}}{1 - \left( \frac{1-\frac{\gamma_{EF}\mu}{4}}{1-\frac{\gamma_{EF}\mu}{2}} \right)^{-t-1} q^{-t-1}} \frac{\varepsilon^2}{4\log 1/\delta} \sum_{k=0}^{t} \frac{n(\gamma_k)}{\gamma_k} q^k (1 + \varepsilon_k) \right]$$

Use the fact, that $\frac{n(\gamma_k)}{\gamma_k} = \frac{a\gamma_k + b\gamma_k^2 + c\gamma_k^3}{\gamma_k} = a + b\gamma_k + c\gamma_k^2 = m(\gamma_k) \leq m(\gamma_t)$ for $k \leq t$. Then,

$$V^{t+1} \leq \prod_{k=0}^{t} \left( 1 - \frac{\gamma_k \mu}{2} \right) \cdot \overline{V}^0 \cdot \left[ 1 + \frac{Ad/(2n)}{1 - \gamma_t \mu/2} \frac{1 - \left( \frac{1-\frac{\gamma_{EF}\mu}{4}}{1-\frac{\gamma_{EF}\mu}{2}} \right)^{-1} q^{-1}}{1 - \left( \frac{1-\frac{\gamma_{EF}\mu}{4}}{1-\frac{\gamma_{EF}\mu}{2}} \right)^{-t-1} q^{-t-1}} \frac{\varepsilon^2}{4\log 1/\delta} m(\gamma_t) \sum_{k=0}^{t} q^k (1 + \varepsilon_k) \right]$$

$$\leq \prod_{k=0}^{t} \left( 1 - \frac{\gamma_k \mu}{2} \right) \cdot \overline{V}^0 \cdot \left[ 1 + \frac{Ad/(2n)}{1 - \gamma_t \mu/2} \frac{1 - \left( \frac{1-\frac{\gamma_{EF}\mu}{4}}{1-\frac{\gamma_{EF}\mu}{2}} \right)^{-1} q^{-1}}{1 - \left( \frac{1-\frac{\gamma_{EF}\mu}{4}}{1-\frac{\gamma_{EF}\mu}{2}} \right)^{-t-1} q^{-t-1}} \frac{\varepsilon^2}{4\log 1/\delta} m(\gamma_t) \left( 2\sum_{k=0}^{t} q^k + \beta \sum_{k=0}^{t} kq^k \right) \right]$$

$$\leq \prod_{k=0}^{t} \left( 1 - \frac{\gamma_k \mu}{2} \right) \cdot \overline{V}^0 \cdot \left[ 1 + \frac{Ad/(2n)}{1 - \gamma_t \mu/2} \frac{1 - \left( \frac{1-\frac{\gamma_{EF}\mu}{4}}{1-\frac{\gamma_{EF}\mu}{2}} \right)^{-1} q^{-1}}{1 - \left( \frac{1-\frac{\gamma_{EF}\mu}{4}}{1-\frac{\gamma_{EF}\mu}{2}} \right)^{-t-1} q^{-t-1}} \frac{\varepsilon^2}{4\log 1/\delta} m(\gamma_t) \left( 2\sum_{k=0}^{\infty} q^k + \beta \sum_{k=0}^{\infty} kq^k \right) \right]$$

$$\leq \prod_{k=0}^{t} \left( 1 - \frac{\gamma_k \mu}{2} \right) \cdot \overline{V}^0 \cdot \left[ 1 + \frac{Ad/(2n)}{1 - \gamma_t \mu/2} \frac{1 - \left( \frac{1-\frac{\gamma_{EF}\mu}{4}}{1-\frac{\gamma_{EF}\mu}{2}} \right)^{-1} q^{-1}}{1 - \left( \frac{1-\frac{\gamma_{EF}\mu}{4}}{1-\frac{\gamma_{EF}\mu}{2}} \right)^{-t-1} q^{-t-1}} \frac{\varepsilon^2}{4\log 1/\delta} m(\gamma_t) \left( \frac{2}{1-q} + \frac{\beta q}{(1-q)^2} \right) \right].$$

The rest of the proof is similar With various $q$ we can influence the privacy budget per iteration.

# E  DESCENT LEMMA 2

Now we will derive the decent lemma for the Algorithm 2.

From $L$-smoothness we obtain

$$f(x^{t+1}) - f_* \leq f(x^t) - f_* - \frac{\gamma_t}{2}\|\nabla f(x^t)\|^2 - \frac{\gamma_t}{2}\|\widetilde{g}^t\|^2 + \frac{\gamma_t}{2}\|\nabla f(x^t) - \widetilde{g}^t\|^2$$

$$\leq f(x^t) - f_* - \frac{\gamma_t}{2}\|\nabla f(x^t)\|^2 - \frac{\gamma_t}{2}\|\widetilde{g}^t\|^2 + \gamma_t\|\nabla f(x^t) - g^t\|^2 + \gamma_t\|g^t - \widetilde{g}^t\|^2.$$

For the EF part the derivation stays the same:

$$\|\nabla f(x^{t+1}) - g^{t+1}\|^2 \leq \frac{1}{n}\sum_{i=1}^n \|\nabla f_i(x^{t+1}) - g_i^{t+1}\|^2 \leq \left(1 - \frac{\alpha}{2}\right)\frac{1}{n}\sum_{i=1}^n \|\nabla f_i(x^t) - g_i^t\|^2 + \left(1 + \frac{2}{\alpha}\right)\gamma_t^2 L^2 \|\widetilde{g}^t\|^2.$$

Our interest is in the second term, where we can obtain following:

$$\gamma_t\|g^t - \widetilde{g}^t\|^2 \leq \frac{\gamma_t}{n}\sum_{i=1}^n \left\|\mathcal{C}\left(\nabla f_i(x^t) - g_i^{t-1}\right) - \mathcal{C}\left(\nabla f_i(x^t) - g_i^{t-1} + \eta_{i,t}\right)\right\|^2$$

$$\leq \frac{2\gamma_t}{n}\sum_{i=1}^n \left\|\mathcal{C}\left(\nabla f_i(x^t) - g_i^{t-1}\right)\right\|^2 + \left\|\mathcal{C}\left(\nabla f_i(x^t) - g_i^{t-1} + \eta_{i,t}\right)\right\|^2$$

$$\leq \frac{2\gamma_t\left(1 + \sqrt{1 - \alpha}\right)^2}{n}\sum_{i=1}^n \left\|\nabla f_i(x^t) - g_i^{t-1}\right\|^2 + \left\|\nabla f_i(x^t) - g_i^{t-1} + \eta_{i,t}\right\|^2$$

$$\leq \frac{\gamma_t\left(1 + \sqrt{1 - \alpha}\right)^2}{n}\sum_{i=1}^n 4\left\|\nabla f_i(x^t) - g_i^{t-1}\right\|^2 + 2\left\|\eta_{i,t}\right\|^2$$

Similarly, we obtain the same asymptotical rates, as above

# F  ADDITIONAL NUMERICAL EXPERIMENTS

We compare Algorithm 2 to 1 and obtain, that results are similar, therefore we may significantly reduce the number of bits sent.  All experiments are done on A100.

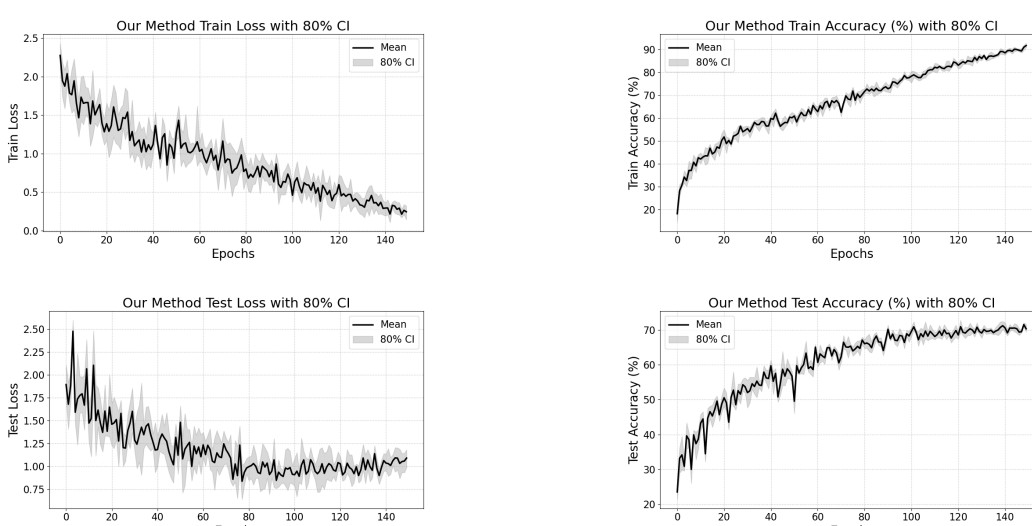

Figure 5: Compression before adding noise

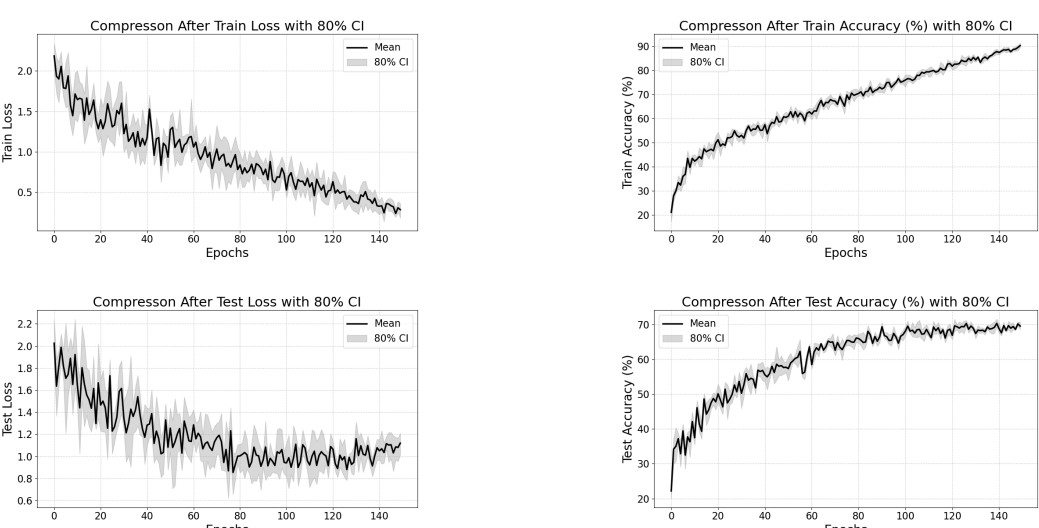

Figure 6: Compression after adding noise

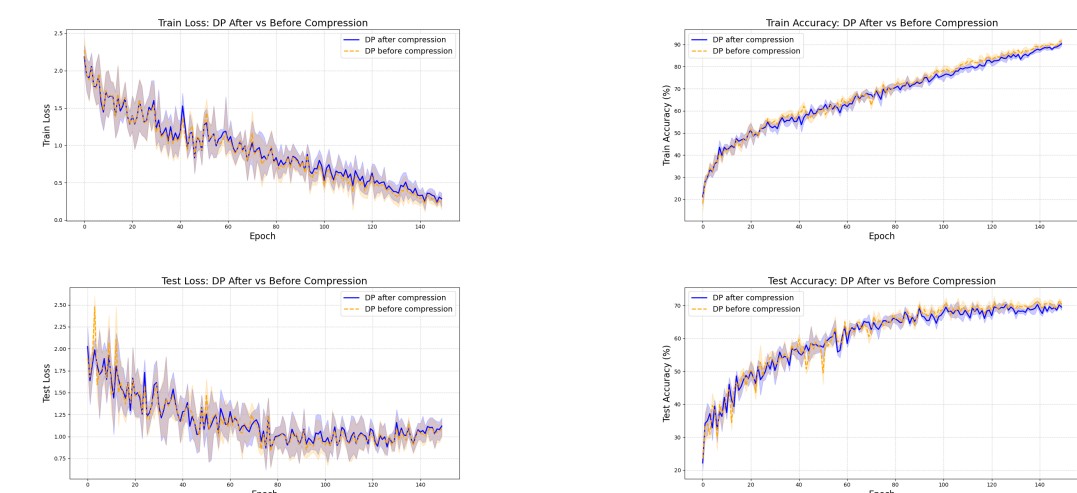

Figure 7: Comparison of compressing before and after adding noise

# G  DECLARATION OF LLM USAGE

We employed Large Language Models to improve the clarity and style of the text.

