# OpenReview forum: "Diminishing Noise Maintains Differential Privacy and Enhances Convergence"
_ICLR.cc/2026/Conference — Submitted to ICLR 2026_

### Official Review · Reviewer_azNw · 2025-10-19

**Soundness:** 2
**Presentation:** 2
**Contribution:** 2
**Rating:** 2
**Confidence:** 3

**Summary:**

This paper proposes a DP optimization algorithm named DPd-EF21 that uses diminishing noise variance to achieve linear convergence while maintaining privacy guarantees. The key contribution is adjusting noise variance proportionally to a Lyapunov function throughout training iterations and enabling convergence. The authors provide a theoretical convergence analysis using concentrated differential privacy (zCDP) and implement the approach on the CIFAR-10 dataset.

**Strengths:**

1. The diminishing noise approach is creative and addresses a key limitation of constant-variance DP methods.
2. The use of zCDP with adaptive composition is more generalizable and allows for varying privacy budgets per iteration.
3. Combining biased compressors with DP (rather than just clipping) maintains communication efficiency, which is important for practical federated learning.

**Weaknesses:**

1. Only one dataset CIFAR-10 is tested. The claims about convergence and privacy-utility tradeoffs need validation across diverse datasets and tasks. Also, Fig. 2-3 shows a comparison for DP-Clip21 and Clip21-SGD2M, which lacks comparisons with other recent DP methods (e.g., DP-SGD variants).
2. The "warming-up DP regime" vs. "main EF21 regime" is mentioned but not characterized precisely.
3. The paper claims that varying privacy budgets with more protection in later iterations is beneficial, but this seems counterintuitive. Later iterations contain more information about the optimum and potentially more sensitive information.
4. No discussion or experiments on problems that don't satisfy PL conditions.
5. No empirical study of how the compression parameter $\alpha$ affects the privacy-utility tradeoff.

**Questions:**

1. How does performance degrade when the PL condition is not satisfied or only approximately holds?
2. How does the method compare to DP-SGD with momentum or other adaptive DP methods not based on error feedback? Where is the comparison to vanilla EF21 without privacy?
3. How was $\sigma_0$ selected for your method vs. constant $\sigma$ for baselines? Was the learning rate tuned separately for each privacy level?
4. In Figure 4, the differences between Top-K levels are small. What drives the choice of sparsification level in practice?
5. What is the computational overhead of tracking Lyapunov functions for adaptive noise scheduling?
6. How many random seeds were used? Are these over $\geq 5$ seeds as recommended for DP experiments?
7. Fig. 4 varies Top-K levels but doesn't show how this interacts with privacy. Can you provide privacy-utility curves for different $\alpha$ values? And convergence speed vs. $\alpha$?
8. How do the results transfer to larger datasets where privacy is more relevant? Or in highly non-convex problems?

---

### Official Review · Reviewer_nrXT · 2025-10-29

**Soundness:** 4
**Presentation:** 3
**Contribution:** 2
**Rating:** 4
**Confidence:** 3

**Summary:**

This paper studies differentially private (DP) training under communication constraints by combining error feedback compression with a diminishing noise variance schedule. The authors propose a DP version of EF21 that uses biased gradient compressors and gradually reduces the added Gaussian noise as training progresses. The key idea is that error-feedback mechanisms produce ever-smaller update magnitudes over time, so the noise needed to preserve privacy can also be decayed proportionally without violating DP. Under the PL condition, the method is proven to achieve linear convergence to the true optimum, rather than only to a noise-dependent neighborhood as in standard DP-SGD. Experiments on CIFAR-10 demonstrate that the proposed diminishing-noise EF21 algorithm attains similar or slightly better accuracy than prior DP optimizers (DP-Clip21 and Clip21-SGD2M) at a given privacy level, while sending fewer bits per round via compression..

**Strengths:**

(1) By introducing a diminishing Gaussian noise variance tied to the algorithm’s progress, the authors address the known issue of noise-induced convergence slowdown. This idea allows the model to eventually converge to the true optimum (under PL), overcoming the inherent accuracy limit of standard DP-SGD which uses fixed noise.
(2) The paper provides rigorous analysis, including privacy proofs using zCDP and convergence proofs with linear rates under PL condition. The use of a Lyapunov function in the proofs is clearly outlined

**Weaknesses:**

(1) The contribution of this work is marginal to me. In essence, the authors revisit two known ideas: error-feedback compression for efficient distributed optimization (as in EF21), and noise decay schedules for DP, and combine them in a single algorithm. Both ingredients are well-studied individually. Meanwhile, the notion of scaling the DP noise to the magnitude of updates has been floated in prior literature (the authors themselves note that most DP methods use constant noise even though theory suggests smaller noise could suffice as updates diminish). The ideas of adaptive noise scaling and even optimal noise scheduling are not new (e.g., [1] and [2]).

(2) The empirical contribution is limited. Experiments are confined to CIFAR-10 with a simple model and only two closely related baselines (DP-Clip21 and Clip21-SGD2M). Important comparisons such as standard DP-SGD and other recent methods employing noise scheduling are missing. Moreover, the reported improvements are marginal and not statistically convincing.
(3) I have concerns about the practical applicability of the proposed algorithm. The paper proposes diminishing the noise variance as the Lyapunov term $V_t$ decreases. In practice, $V_t$ is not known in advance. How should one set or tune the noise schedule on a new problem to ensure both privacy and convergence?

[1] Fu J, Chen Z, Han X. Adap dp-fl: Differentially private federated learning with adaptive noise[C]//2022 IEEE international conference on trust, security and privacy in computing and communications (TrustCom). IEEE, 2022: 656-663.
[2] Geng Q, Viswanath P. The optimal noise-adding mechanism in differential privacy[J]. IEEE Transactions on Information Theory, 2015, 62(2): 925-951.

**Questions:**

(1) The authors assert their method preserves “communication efficiency” which prior DP variants lacked, but this could have been demonstrated more clearly (perhaps by measuring actual bandwidth or scalability).
(2)  How sensitive are the results to the choice of noise schedule? For example, the experiments used a specific geometric decay and a Lyapunov-proportional schedule. Would any monotonic decay of noise work, or does it need to closely track the theoretical $V_t$?

---

### Official Review · Reviewer_c7Yy · 2025-10-30

**Soundness:** 1
**Presentation:** 1
**Contribution:** 1
**Rating:** 0
**Confidence:** 4

**Summary:**

The paper attempts to add noise with diminishing variance in a FL setup with compressed feedback.

**Strengths:**

The idea of a noise scheduler in the context of FL with compressed feedback is interesting. However there seem to serious technical errors in the paper. Please see the detailed review below

**Weaknesses:**

Please see the detailed review below

**Questions:**

Major concerns:
There seems to be a fundamental mistake in the treatment of sensitivity in this paper.

In lines 273 and 889, the sensitivity has been incorrectly calculated: while $D$ has been treated fixed, and the maximum has been obtained over $ \{ D’ | D’ \sim D \} $, the actual definition of sensitivity involves supremum over every pair of neighbouring datasets, i.e., over $\{ (D,D’) | D’ \sim D \}$. This error propagates throughout the theoretical development, and the subsequent derivation of the noise variance and the convergence analysis are invalid.

The sensitivity bound provided in the manuscript would be valid if the authors had considered local sensitivity. But then, calibrating noise with respect to local sensitivity would not offer proper privacy protection [1, Sec. 1.3], which is essentially the reason why it has not been used in practice. Specifically, measuring sensitivity for the dataset at hand would essentially make the noise parameters depend on that dataset (here through $V^t$), which violates the purpose. Local sensitivity measure must be smoothed [1] in order to be useful. However, such smoothening has not been considered by the authors, and they did not provide proper justification for the use of local sensitivity.

In fact, in line 330, the authors state that their algorithm converges to the exact solution. However, since the optimization landscape is shaped by the dataset, any technique that provides exact convergence will not be private.


Other comments:
    Writing in inconsistent. Examples: “183: Let us discuss these claims.” No claims have been made yet. Suddenly mentioning   approximate DP without context in Line 191

 Define and expand PL condition when it is first used

 The main theorem claims convergence and privacy. The privacy guarantee simply quotes composition theorems and it is not explicitly derived

  The main sections should explicitly refer to relevant appendix sections for proof of ready readability

  Empirical results are restricted, only results for CIFAR-10 is provided. Including different datasets and models is necessary to strengthen the empirical validation

 The PL condition is a strong condition and I am not clear why is should hold for the CIFAR10 setup and how you are able to find the parameter $\mu$. More concretely how are you estimating the parameters you require for your noise schedules in line 410- $V_t$ itself will not be deterministic once you add noise and how will you determine $\mu$?

  Minimum details (such as the model being trained on, no. of seeds the results are averaged over, optimization method, etc.) should be included in the main paper

  The experimental evaluation claims to provide results on privacy-utility tradeoff but all results are for matched privacy. Typically, for such studies , the privacy budget epsilon is varied and its effect on the system’s performance is studied



[1] K. Nissim, S. Raskhodnikova, and A. Smith, "Smooth sensitivity and sampling in private data analysis," Proceedings of the thirty-ninth annual ACM symposium on Theory of computing, 2007. https://dl.acm.org/doi/10.1145/1250790.1250803

---

### Official Review · Reviewer_UPEB · 2025-11-01

**Soundness:** 2
**Presentation:** 1
**Contribution:** 2
**Rating:** 2
**Confidence:** 4

**Summary:**

The paper proposes DPd-EF21, a differentially private variant of the EF21 error-feedback algorithm for distributed optimization. The key idea is to use diminishing Gaussian noise whose variance decreases with the magnitude of gradient updates, aiming to maintain differential privacy (DP) while achieving linear convergence under the Polyak–Łojasiewicz (PL) condition. Theoretical results claim convergence to the exact optimum (Theorem 1) and valid DP guarantees (Theorem 2). Experiments on CIFAR-10 provide limited evidence.

**Strengths:**

* Introduces the concept of diminishing DP noise in gradient compression, connecting privacy and error-feedback mechanisms.
* Attempts to theoretically show linear convergence under DP (Theorem 1) and ensure privacy consistency via zCDP (Theorem 2).
* Integrates analysis of biased compressors, extending prior EF21 results.

**Weaknesses:**

- **Unclear research question:** The paper never clearly states the specific gap it fills beyond restating known EF21 and DP mechanisms.
- **Poor exposition:** Writing quality severely hinders understanding. Equations are dense, unmotivated, and often unexplained.  No clear comparison to prior theoretical results to justify the advantage in convergence. Not theoretical justification on the privacy cost for the algorithm
- **Insufficient comparison:** Only compares with DP-Clip21 and Clip21-SGD2M, ignoring many cited baselines such as adaptive clipping or other DP compression methods.
- **No communication analysis:** Despite claims of communication efficiency, no quantitative results on bandwidth or transmitted bits are provided.
- **Single dataset and limited experiments:** Only CIFAR-10 is tested, with no exploration of non-IID heterogeneity, larger benchmarks, or ablations.

**Questions:**

1. What concrete problem does the paper solve beyond constant-noise DP-SGD? Please specify the novelty more precisely.
2. How is the per-iteration privacy composition handled when the noise variance changes dynamically?
3. Can you empirically validate the assumption that sensitivity (and thus noise variance) diminishes with iteration?

---

### Meta-Review · Area_Chair_SZSe · 2025-12-12

**Summary:**

All four reviewers find this paper not suited for publication in its current state. The reviewers addressed several issues, but the authors did not provide any responses. Hence, I recommend rejecting the paper.

**Reviewer Concerns:**

The reviewers addressed several issues, but the authors did not provide any responses.

**Reviewer Scores:**

I have no reason to believe the reviewers would have increased the scores without any response from the authors.

---

### Decision · Program_Chairs · 2026-01-26

Reject